# Historical Graph Management in Dynamic Environments

**Kyoungsoo Bok [1], Gihoon Kim [2], Jongtae Lim [2] and Jaesoo Yoo [2],\***

[1] Department of SW Convergence Technology, Wonkwang University, Iksandae 460, Iksan, Jeonbuk 54538, Korea; ksbok@wku.ac.kr
[2] Department of Information and Communication Engineering, Chungbuk National University, Chungdae-ro 1, Seowon-Gu, Cheongju, Chungbuk 28644, Korea; argenston@naver.com (G.K.); jtlim@chungbuk.ac.kr (J.L.)
[*] Correspondence: yjs@chungbuk.ac.kr; Tel.: +82-43-261-3230

**Abstract:** Since dynamic graph data continuously change over time, it is necessary to manage historical data for accessing a snapshot graph at a specific time. In this paper, we propose a new historical graph management scheme that consists of an intersection snapshot and a delta snapshot to enhance storage utilization and historical graph accessibility. The proposed scheme constantly detects graph changes and calculates a common subgraph ratio between historical graphs over time. If the common subgraph ratio is lower than a threshold value, the intersection snapshot stores the common subgraphs within a time interval. A delta snapshot stores the subgraphs that are not contained in the intersection snapshot. Several delta snapshots are connected to the intersection snapshot to maintain the modified subgraph over time. The efficiency of storage space is improved by managing common subgraphs stored in the intersection snapshot. Furthermore, the intersection and delta snapshots can be connected to search a graph at a specific time. We show the superiority of the proposed scheme through various performance evaluations.

**Keywords:** dynamic graph; historical data; storage utilization; common subgraph; intersection snapshot; delta snapshot

## 1. Introduction

Graph data have been used to represent the interactions or relationships between objects through vertexes and edges [1–3]. In recent years, new information has been constantly generated and existing information changes through social networks, citation networks, and the Internet of Things (IoT) [4–8]. The graph data that represent such information generate dynamic graphs that continuously change through various update operations [9–11]. Dynamic graphs with continuous changes generate a large amount of historical data [12–15]. To track the history of changes in graphs or to search for graphs at a specific time in the past in a dynamic environment, a historical graph is required to manage the continuous changes in the vertices and edges that make up the graph [16–20]. When a snapshot graph $G_n = (V_n, E_n)$ exists in a specific time $n$, the history graph $G = \langle G_1, G_2, \ldots, G_{current} \rangle$ consists of all the changed graphs $G_n$ from the past to the present. Here, $V_n$ and $E_n$ are sets of vertices and edges present at the time $n$. That is, historical graphs do not store only the graphs at a specific point in time or the final graphs that have changed, but they also store changes in the vertices and edges that have changed continuously in the initial graph so that we can analyze the graph changes or view the graph's status at a specific time.

A typical graph is used to express relationships or interactions between objects and to search for or analyze relationships between objects. A historical graph is used to search for vertices and

edges that exist at specific points or intervals or to analyze changes in a graph over time, since they store changes in the graph that have changed from the past to the present. For example, if we model relationships between virus-infected people or connections as a graph, we can provide a confirmed spread path by creating a web map of the COVID-19 diffusion path, which has become a recent social issue, and provide time-to-time status of the spread of the virus along the path of the confirmed person's movement [21,22]. In addition, if we model relationships and information delivery among users on social networks as graphs, we can search for or analyze queries such as "What has become of the spread of information over time?", "What is the 2020 issue compared to 2019?", and "Who is the most influential user in 2019?" [16,17]. DataBase systems and Logic Programming (DBLP) and the citation network can model the relationships between the papers and the authors as a graph, identify the changes in the cited paper as time changes, and provide researchers with changes in the research topic through analysis of the relationship between the author and the paper from the past to the present [5,6].

Facebook, a leading social network service, has 1.39 billion active users and Twitter has 228 million active users as of March 2015 [9]. In this environment, storing and managing the sharing information on social network services over time as historical graphs take a lot of storage space, and it takes a lot of time to search or analyze graphs that exist in a specific time or a specific time interval [23–28]. The graph storage structure for storing graphs individually over each time has been proposed. The Spatio-Temporal Interaction Networks and Graphs Extensible Presentation (STINGER) uses a linked list-based structure to store large amounts of graph [24]. Degree Aware Robin Hood Hashing (DegAwareRHH) determine where data is stored by using the robin-hood hashing technique [25]. In the existing graph storage structure, a large amount of common subgraphs will be stored if graphs are maintained for the entire time domain even if there are some subgraph changes. The common subgraph storage for equivalent information causes storage space wastage. For example, a relationship graph for a user in a social network only has changes in some friend relationships rather than the entire friend relationship.

Historical graph management schemes that access a large amount of existing graph data and track the history of graph data through a scalable structure have been proposed [29–32]. Historical graph management schemes are divided into a copy method and a log method [28]. The copy method maintains all time-based graphs to quickly search for subgraphs that exist at a specific time. However, since most applications only change part of the graph, the unchanged subgraph is stored in duplicate, which uses a lot of storage space. To solve this problem of storage space, the log method was proposed to store mainly changed subgraphs. A compact representation of graph snapshots called Version Graph preserves the subgraph for a specified time period after setting the preservation interval [29]. Since the Version Graph does not store all graphs but only the change history, it has good space efficiency in terms of the spatial perspective. However, it has the drawback of slow access speed, because it has to access the change history sequentially from the first graph to check the graph data at the required time period. A distributed hierarchical index structure called DeltaGraph increases the efficiency of storage space by utilizing the graph change history to reduce unnecessary graph storage [30]. However, graph access was inefficient due to the sequential access to the change history. The existing schemes exhibited a trade-off relationship between two approaches as they showed poor performance at their weak point in terms of storage and access. Thus, a scheme that could take both methods into account appropriately for graph storage and thus access historical graph more efficiently was necessary, in consideration of all circumstances.

In this paper, we propose an efficient historical graph management scheme to reduce the storage space and improve the processing time for snapshot queries. The proposed scheme stores a common subgraph in an intersection snapshot and a modified subgraph over time in a delta snapshot by using the feature that the parts of the graph change over time. The intersection snapshot and delta snapshots are connected to enable the accessing of graphs at a specific time. The proposed scheme can increase the space efficiency because it does not store duplicate historical graph to solve the problem of

space consumption. A historical graph is represented by the modified Compressed Sparse Row (CSR) technique to reduce the graph storage space. The proposed scheme has the following characteristics.

- It integrates and manages all changed information on a single graph using a provenance model that manages the history of data changes.
- It analyzes changes in graphs managed through the provenance model and generates an intersection snapshot and a delta snapshot if the change rate is above the threshold.
- The intersection snapshot stores common subgraphs for reducing the duplicate storage and leading to a reduction in storage space.
- The delta snapshot stores subgraphs at each time that is not included in the intersection snapshot to retrieve the changed subgraph.
- It stores the intersection snapshot and delta snapshot to reduce storage space on the history graph using a modified Compressed Sparse Row (CSR).

The rest of this paper is organized as follows. Section 2 describes related works, the proposed historical graph management scheme is described in Section 3, and the performance evaluation of the proposed scheme is presented in Section 4 through comparison with existing schemes. Section 5 presents the conclusion.

## 2. Related Work

STINGER is a graph storage structure based on linked lists of blocks to support fast insertions, deletions, and updates [24]. An edge is represented as a tuple of neighbor vertex ID, type, weight, and two timestamps. All edges in a given block have the same edge type. Each vertex is stored in the Logical Vertex Array (LVA), in which a pointer to a block that stores how several edges connect to a corresponding vertex is maintained. A pointer to a block has location information about the block for edge storage, and multiple blocks are connected as a form of connection list. Thus, it can store a large amount of graph in a scalable manner. STINGER employs an edge-type array for fast access to each edge, and the edge-type array (ETA) is used to minimize the sequential access, which is a drawback of the connection list due to the inherent structure when accessing graphs based on edges rather than vertices. The edge-type array (ETA) is a secondary index that points to all edge blocks of a given type. The ETA contains a pointer to indicate the location information of the block to which a particular edge belongs according to the type of edge. The STINGER supports read-only queries of vertices and edges, as well as the insertion and deletion of both vertices and edges.

DegAwareRHH is a distributed graph data storage structure for a dynamic graph [25]. DegAwareRHH is the adjacency list structure using robin-hood hashing. Each edge is stored in two types of tables such as a low-degree table and a middle-high degree table. The low-degree table stores edges in a single table. The middle-high-degree table consists of a vertex table and edge chunks. The vertex table stores source vertices' information. Adjacency edges are stored into edge chunks, and the vertex table holds pointers to edge chunks. Storing a large amount of graph data efficiently in a distributed manner requires that a graph storage location is determined to maintain data locality by utilizing an existing robin hood hashing scheme that stores graphs in a manner that considers graph locality when new graphs are added to the page on which existing data are stored. If the added graph is to be stored at the location of the existing graph, the existing graph location is moved. Vertices with a large number of connected edges require a large space to represent each edge.

A Linked-node analytics using Large Multiversioned Arrays (LLAMA) was proposed to resolve the waste of storage space in existing structures and improve scalability for the storage of a dynamic graph [33]. LLAMA is a scalable graph storage structure that utilizes a multiversion array. A snapshot is made once a graph changes to store change histories. It has the advantage that it can cope with constant changes, which is a characteristic of dynamic graphs. A scalable structure is created by dividing a single snapshot into a vertex table and edge table in the configuration. The vertex table maintains the snapshot ID of where the out-going vertex is located, the location information of edges

connected to the out-going vertex, and the number of connected vertices; the edge table maintains the in-coming vertex ID to express the out-going and in-coming vertices of edges. A single dynamic graph is expressed by maintaining the aforementioned structure of the snapshot in many versions. Once data change according to the changes in the dynamic graph, a new snapshot is made. For data without changes, the edge information in the edge table contains pointer information that points to a room of the edge table in the previous snapshot. Scalable graph data can be stored efficiently in the storage space by utilizing multiversion snapshots.

Version Graph is a graph data structure proposed for accessing the past histories of dynamic graphs [29]. Information about all vertices and edges that were added to the graph over time is stored and the lifespan of each vertex and edge is recorded in the graph to check whether vertices and edges exist in the graph at a specific time. The lifespan is sets of time intervals indicating vertices and edges is deleted and then reinserted at snapshot graphs. A Strongly Connected Component (SCC) is employed to search changes over time. Version Graph maintains posting lists with information about node membership in SCCs. Version Graph minimize the size of posting lists through an appropriate assignment of identifiers to SCCs. Thus, Version Graph can be accessible to subgraphs that change over time efficiently; it can maintain all historical graph to facilitate fast access to past graphs.

DeltaGraph is a graph data management system that stores the change histories of how graph data change over time. It manages past histories using a hierarchical structure [30]. At the end node, graph data with respect to time are positioned, and intermediate nodes that permit access to each graph data are laid at the upper nodes. Graphs at specific times are configured as snapshots to maintain the changes that occur between snapshots through the event list. DeltaGraph is represented as a tree structure to find a snapshot that will be a reference to starting sequential access by tracking events. End nodes can be accessed by the shortest accessible distance from the super-root in the uppermost end to the end node, and they are calculated to access an end node. The super-root is a virtual access node; it shows a blank graph without vertices or edges. A snapshot is chosen that is a reference to the start of a sequential search for events using a tree structure. Starting from the chosen snapshot, histories that change through events at particular times are updated in snapshots to access the graph data at the preferred time. DeltaGraph accesses an end node by calculating the shortest path to a particular time so that it can access the graph at that particular time, and it performs sequential access by tracking histories about graph states between end nodes.

Various approaches have been proposed to store and search for continuous changes in graphs over time. The existing graph storage structures can store a large amount of graph data by considering data scalability [24,25,33]. STINGER is a large graph storage structure that takes into account scalability. Access to vertices and edges is available through LVA and ETA, but past histories of graphs are not accessible. Using Hash, DegAwareRHH can efficiently distribute and store large amount of graphs by maintaining locality and classifying vertices according to the number of edges. LLAMA manages dynamic graphs with multiple versions of arrays using snapshots. Access to each snapshot requires a lot of processing time because it is phased from the first change history. Therefore, existing graph storage structures cannot be analyzed, since they do not maintain past graphs. The historic graph management schemes have been proposed to search for changed past history [29,30]. Version Graph provides efficient access to subgraphs that have changed over time and maintains all historical data for quick access to past graphs. However, a lot of storage space is needed because it maintains the structure of all graphs that have changed in the past. DeltaGraph accesses the leaf nodes by calculating the shortest path from the tree structure to that time point in order to access the graph at that point. Since the leaf node includes the event information including vertices or edges are created or deleted, the process of analyzing events and creating actual graphs while sequentially approaching the leaf node is necessary to verify the graph at a specific time. It takes a lot of time to access a graph at a specific time because the actual graph is generated using events while accessing the leaf nodes sequentially. The historical graph management schemes have to access a large number of histories sequentially to access past graphs or require a large amount of storage space to maintain all graphs,

both of which are problematic. The problems of the two schemes are that they are in a trade-off relationship. Thus, it is necessary to have a storage management structure that can store graph data efficiently and access past graphs effectively for historical graph analysis by taking advantage of two schemes appropriately that are in a trade-off relationship between copy and log-based access. Thus, we proposes an efficient storage management scheme to search historical graphs by combining copy and log-based access methods appropriately through utilizing a storage structure that modifies a CSR technique to reduce the amount of information in graph data and manage historical graphs through the division of snapshots into two layers.

## 3. The Proposed Historical Graph Management Scheme

### 3.1. Overall Structure

Dynamic graphs have much in common with subgraphs, causing wasted space due to duplicate storage. We present a new a historical graph management scheme that minimizes the storage of common subgraphs and efficiently access snapshot graphs. The accessibility of past graphs can be ensured while minimizing common subgraphs by dividing and managing data into intersection and delta snapshots. An Intersection Snapshot (IS) is generated with common subgraphs according to a common subgraph ratio. The common subgraphs are managed in the intersection snapshot in an integrated manner to reduce storage space wastage, and the remaining subgraphs except for the common graphs are stored in the Delta Snapshot (DS). A single snapshot includes tables of out-going and in-coming vertices. An out-going vertex represents an edge between two vertices by the number of vertices connected with an offset that indicates a starting position in the corresponding in-coming vertex table. Furthermore, the CSR technique is utilized to reduce the amount of information in graph data.

Figure 1 shows the proposed structure of the historical graph management. The overall structure of the proposed scheme is configured with an IS at the upper end and DS that consists of changed subgraphs over time at the lower end. The common subgraphs in the temporal graphs are stored in the IS, and the DS increases the storage efficiency by only storing the changed subgraphs in the common subgraph. The time information in a DS is connected to a corresponding ID and maintained for access to snapshots. Figure 1a,b show that $G_1$ and $G_2$ have a common subgraph, where $v_i$ is a vertex and $e_{ij}$ is an edge connecting $v_i$ and $v_j$. The common subgraph $\{e_{12}, e_{24}, e_{13}\}$ is stored in $IS_1$, as shown in the structure of (c) in Figure 1. The subgraph $\{e_{25}, e_{62}\}$ that is not stored in $IS_1$ is stored in $DS_1$ for $G_1$ and the subgraph $\{e_{21}, e_{53}, e_{56}\}$ is stored in $DS_2$ for $G_2$. To support the historical search, $DS_1$ and $DS_2$ are connected to the $IS_1$.

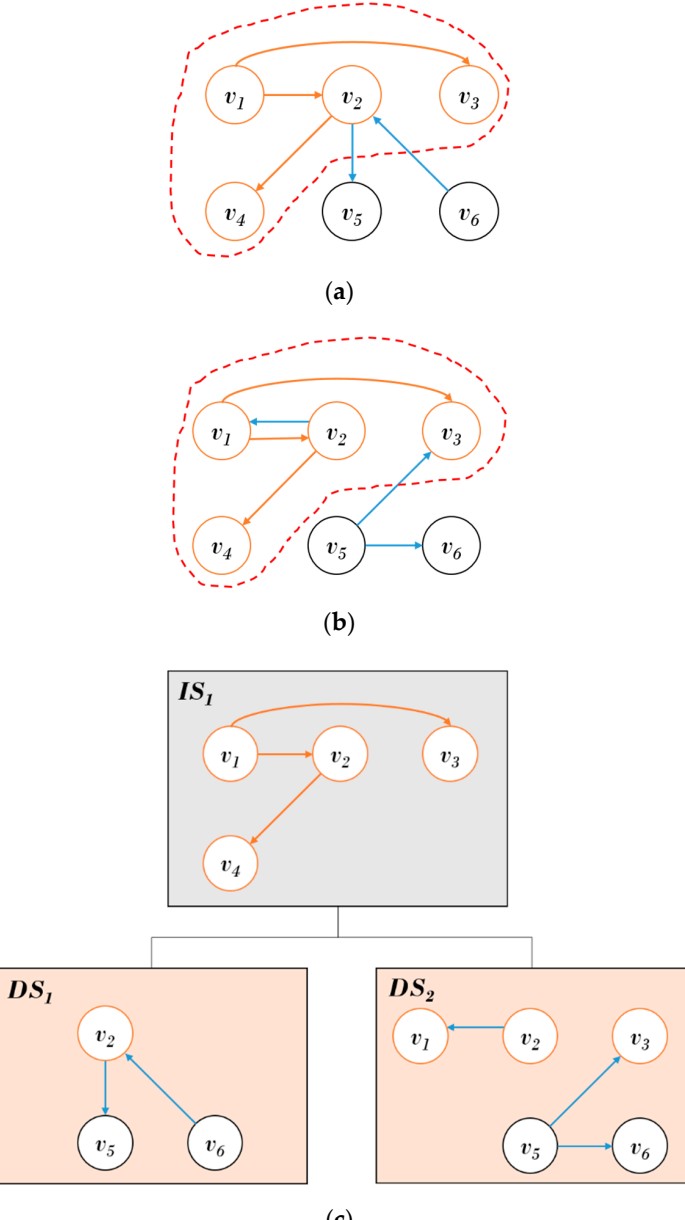

**Figure 1.** Proposed historical graph management structure: (**a**) Graph $G_1$ at time $T_1$; (**b**) Graph $G_2$ at time $T_2$; (**c**) Historical graph.

IS is generated by detecting common subgraphs sequentially by time from the first graph data. IS is generated for which the Common SubGraph Ratio (CSGR) of the historical graphs is larger than the threshold value. The DS is generated to include subgraphs at each time that is not included in the detected common subgraphs in the graph data up until the time generated by the IS and the generated DS is connected to the corresponding IS. The proposed overall structure is constructed by iterating the above process. Figure 2 shows the flow chart to generate the overall snapshot. We calculate a CSGR between consecutive graphs to generate $IS_i$. That is, we calculate $CSGR_{in}$ between $IS_i$ and the next time graph $G_n$. $CSGR_{in}$ is calculated by Equation (1), where $|G_n|$ is the number of edges for $G_n$ and $|IS_i \cap G_n|$ is the number of common edges between $IS_i$ and $G_n$. If $CSGR_{in}$ is larger than the threshold $\varepsilon$, the $IS_i$ is changed. After updating $IS_i$, the comparison with the next time graph is performed iteratively

to change the $IS_i$. If $CSGR_{in}$ between $IS_i$ and new graph $G_n$ is smaller than $\varepsilon$, $IS_i$ is generated as the IS, and the corresponding $DS_n$ are generated as the IS from $G_n$ for $k$ historical graphs.

$$CSGR_{in} = \frac{|IS_i \cap G_n|}{|G_n|}. \tag{1}$$

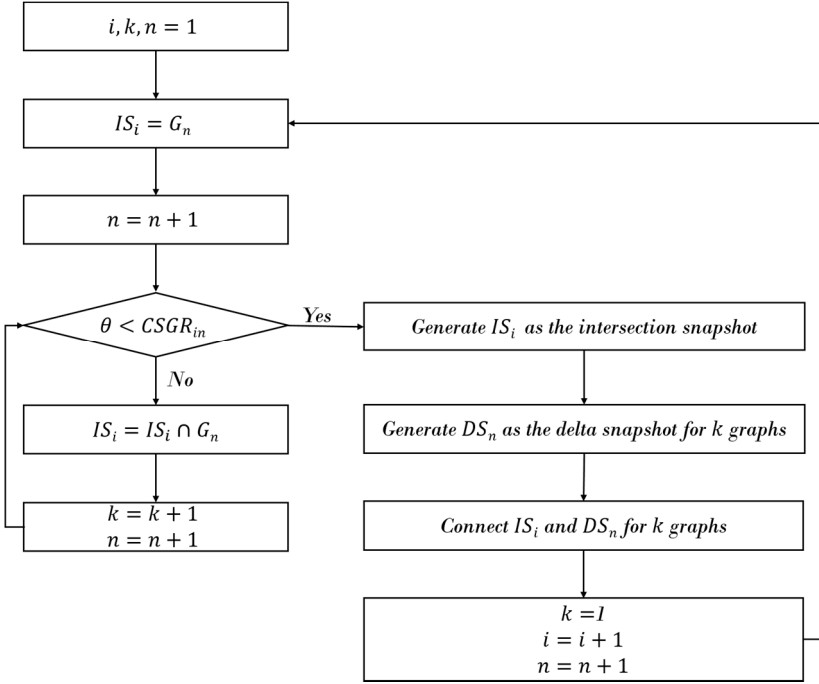

**Figure 2.** The process of generating the overall snapshot.

### 3.2. Intersection Snapshot

Dynamic graphs are likely to have partial changes over time rather than changes to the entire graph, and the subgraphs without changes will be duplicated over time. The common subgraphs waste storage space in an integrated graph requires ISs. The ISs are individually stored to reduce storage space and search performance. The IS only stores common subgraphs within a time interval in contrast with snapshots in existing schemes. Since the generated IS does not store common subgraphs, it can minimize the waste of storage space. Each $IS_i$. stores $\langle TI_i, NS_i, CSG_i \rangle$. Here, $TI_i$ is a time interval $[ST_i, ET_i]$ of $IS_i$, where $ST_i$ and $ET_i$ are the first changed time and last changed time of graph $G_n$ connected to $IS_i$. Equations (2) and (3) are $ST_i$ and $ET_i$, where $T(G_n)$ is a changed time of $G_n$ [29]. $NS_i$ is a the number of historical graphs connected to $IS_i$ and $CSG_i$ is common subgraphs contained in $IS_i$. If a graph $G_n$ is a first graph connected in $IS_i$, $CSG_i$ is calculated by Equation (4) similar to [17], where '∩' is an intersection operation that stores common subgraph of graphs.

$$ST_i = Min_{n=j}^{j+NS_i-1} T(G_n) \tag{2}$$

$$ET_i = Max_{n=j}^{j+NS_i-1} T(G_n) \tag{3}$$

$$CSG_i = \cap_{j=1}^{j+NS_i-1} G_j \tag{4}$$

To generate the IS, the common subgraphs in the graphs over time are sequentially detected. When there are $k$ consecutive graphs, the common subgraphs and the change histories are analyzed by time. If a CSGR is lower than a threshold value after analysis, the IS for $k$ historical graphs is generated. Otherwise, the CSGR for the next historical graph is calculated. We continuously compare

the consecutive graphs, and the Common SubGraph (CSG) and Provenance Information (PI) is stored in a Graph Pattern Table (GPT) over time. The CSG stores common edges among consecutive graphs. In dynamic graphs, vertices and edges are inserted and deleted over time. Originally, provenance are metadata that represent the source information or changing history of data [34–36]. The provenance can be used to track the data changes and usage histories. The proposed scheme should detect the changed subgraph over the time to detect the CSG. Therefore, the proposed scheme uses the provenance to track the update operation of graphs over time. The PI stores the changed information of the current graph through comparison with the previous graph by simplifying the original provenance representation. In PI, each changed information $UI_j$ is represented by $\left(OP_j, OB_j, T_j\right)$, where $OP_j$ is a update operation such as insert ($I$) and delete ($D$), $OB_j$ is a changed object such as vertex and edge, and $T_j$ is a changed time of $OB_j$.

Figure 3 shows the generation procedure of IS. Let us assume that there are three graphs that changed from $T_1$ to $T_3$. Table 1 shows a GPT from $T_1$ to $T_3$. Since the graph $G_1$ is a start graph, the common edges is itself. The graph $G_2$ has five common edges with $G_1$ and is changed by the four update operations compared to $G_1$. Similarly, $G_3$ has five common edges and is changed by the six update operations compared to $G_2$. If the CSGR of $G_4$ is lower than a threshold value, we generate $IS_1$ as shown in Figure 3d.

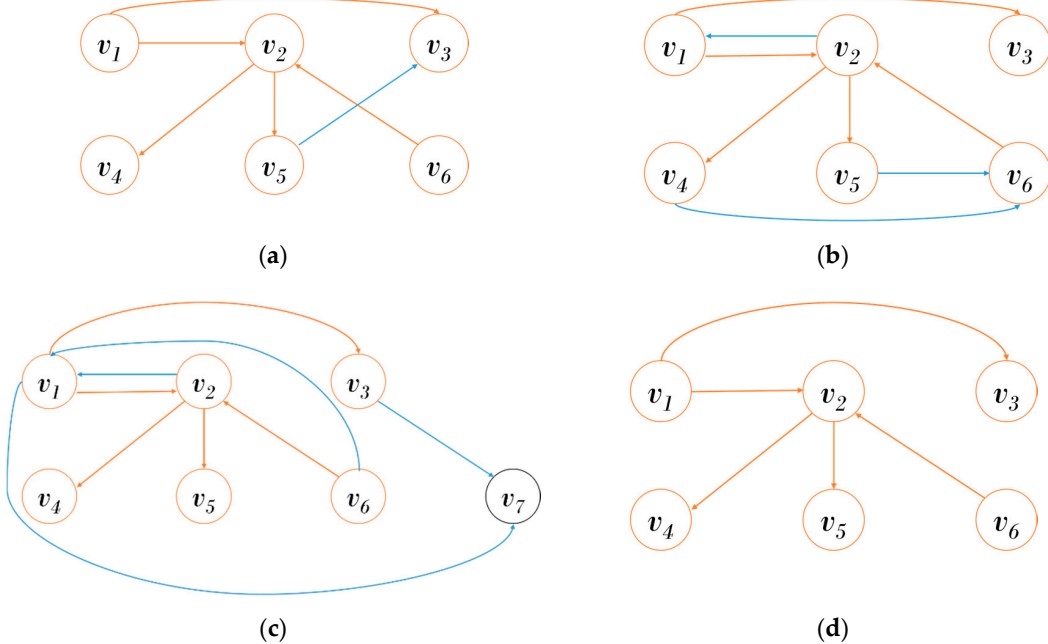

**Figure 3.** Generation of an Intersection Snapshot (IS): (**a**) $G_1$ at time $T_1$; (**b**) $G_2$ at time $T_2$; (**c**) $G_3$ at time $T_3$; (**d**) IS.

**Table 1.** Graph pattern table. PI: Provenance Information.

| Graph | CSG | PI |
|---|---|---|
| $G_1$ | $\{e_{12}, e_{13}, e_{24}, e_{25}, e_{53}, e_{62}\}$ | *Null* |
| $G_2$ | $\{e_{12}, e_{13}, e_{24}, e_{25}, e_{62}\}$ | $\{(I, e_{21}, T_2), (I, e_{46}, T_2), (D, e_{53}, T_2), (I, e_{56}, T_2)\}$ |
| $G_3$ | $\{e_{12}, e_{13}, e_{24}, e_{25}, e_{62}\}$ | $\{(I, e_{37}, T_3), (D, e_{46}, T_3), (D, e_{56}, T_3), (I, v_7, T_3), (I, e_{17}, T_3), (I, e_{61}, T_3)\}$ |

### 3.3. Delta Snapshot

If $k$ historical graphs are stored separately during generating IS, it requires a lot of storage space and search cost. The proposed scheme manages $k$ historical graphs as a provenance graph with the PI to support a historical search. However, dynamic graphs are continuously changed, and then the

size of the provenance graph is continuously increased. Therefore, we generate a DS that stores the subgraphs that are not contained in the IS for the $k$ historical graph and each $DS_n$ is connect to the corresponding $IS_i$. $DS_n$ of a graph $G_n$ manages $(T_n, SG_n)$, where $T_n$ is a changed time of $G_n$ and $SG_n$ is a subgraph of $G_n$ that is not contained in $IS_i$. The provenance graph stores the original graph as well as the changing history. Since $IS_i$ stores a common subgraph $CSG_i$, we first remove $CSG_i$ included in $IS_i$ from the provenance graph to generate $DS_n$. If $G_j$ is the first graph connected in $IS_i$, $DS_n$ is generated by removing all the changed histories. If $G_n$ is not the first graph connected to $IS_i$, $DS_n$ is generated by reflecting the changing history information. $SG_n$ is calculated by Equation (5) similar to [29,30], where $PG_i$ is a provenance graph, $PI_n$ is the provenance information of $SG_n$, '−' is an operation that removes certain subgraphs or withdraws update operations, and '+' is an operation that reflects the update operations.

$$SG_n = \begin{cases} PG_i - CSG_i - \sum_{n=j}^{n+k-1} PI_n & , \; if \; SG_n \; is \; the \; first \; graph \; connected \; to \; IS_j \\ SG_{n+1} + PI_n & , \; otherwise \end{cases} \tag{5}$$

Figure 4 shows the generation procedure of DS. Figure 4a is the provenance graph expressed using the PI shown in Table 1. To generate the $DS_n$ of $G_n$, we first remove a common subgraph $\{e_{12}, e_{13}, e_{24}, e_{25}, e_{62}\}$, as shown in Figure 4b. Since $G_1$ is the first graph connected to $IS_1$, $DS_1$ is generated by removing all PIs, as shown in Figure 4c. Since $G_2$ has performed the four update operations on $G_1$, $DS_2$ is generated by reflecting the update operations as shown in Figure 4d. $DS_3$ is also generated by reflecting the update operations of $G_3$, as shown in Figure 4e.

If the out-going vertex in the stored subgraph is the same as the out-going vertex in the IS, they are connected via the offset information in the table. The connected DS includes additional information about each graph that is not duplicated in the IS. Its internal structure is similar to that of the IS. It has out-going and in-coming vertex tables and time information. The out-going vertex table maintains all out-going vertices in the edge, and the in-coming vertex table has the in-coming vertex information of the added edges. The in-coming vertex of an edge in the IS has the location information of the common graph data using the number of vertices connected to the offset in the in-coming vertex table of an IS.

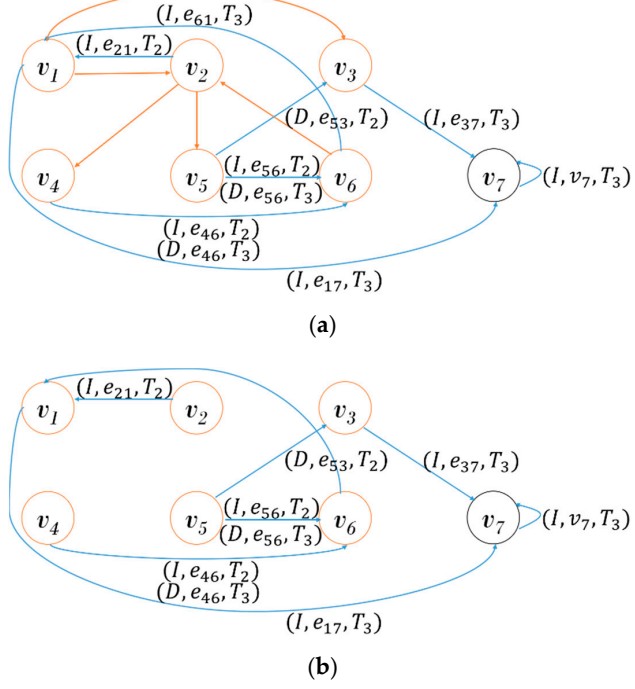

**(a)**

**(b)**

**Figure 4.** *Cont.*

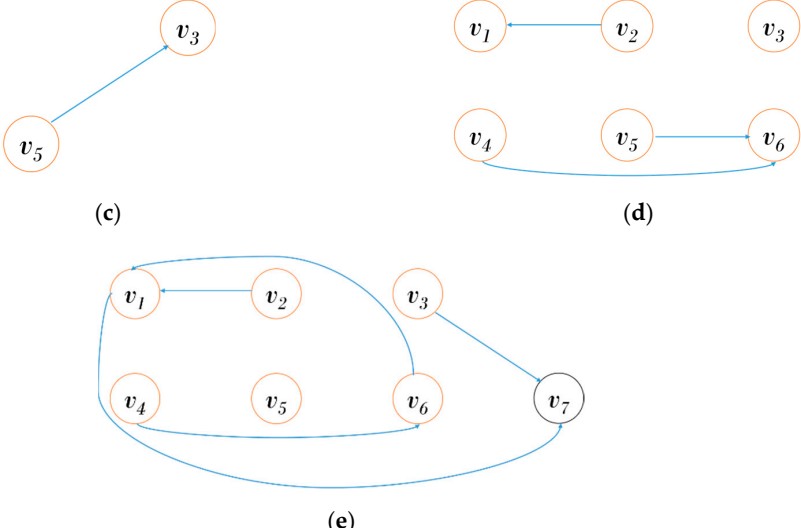

(c)

(d)

(e)

**Figure 4.** Generation of Delta Snapshot (DS): (**a**) Provenance graph; (**b**) Removing Common SubGraph (CSG); (**c**) $DS_1$; (**d**) $DS_2$; (**e**) $DS_3$.

*3.4. Graph Representation*

Generally, the existence of edges that connect vertices is represented by a sparse matrix in graph data. However, if graph data are represented by a sparse matrix, it requires a matrix size that includes the total number of vertices. Furthermore, most values in a matrix whose edges do not exist are represented by zero. Thus, the CSR is used to store the sparse matrix representation of graph data more efficiently [37]. CSR representation does not represent an unnecessary matrix whose edges are not present but connects only existing data according to the order of rows. The proposed scheme modifies the CSR method to connect snapshots. Figure 5a shows the representation of a dynamic graph through a sparse matrix. Most values in the matrix are filled with zero, which represents unnecessarily data. Figure 5b shows the sparse matrix at times $T_1$ and $T_2$ using the CSR representation technique. Each row is represented in the Row Pointer Array (RPA), which indicates a position in the Column Indices Array (CIA). The RPA represents a sparse matrix with location information about the row in which the data actually exist. The space efficiency can be increased by not representing unnecessary space as described above.

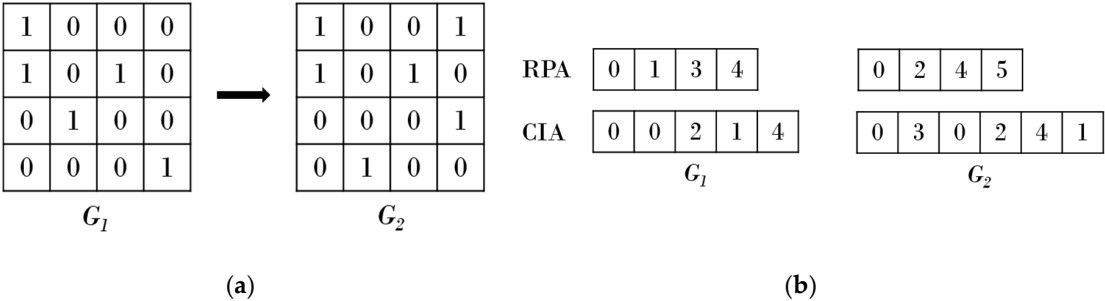

(a)

(b)

**Figure 5.** Compressed Sparse Row (CSR): (**a**) Sparse matrix representation for dynamic graphs; (**b**) Existing CSR representation.

The existing CSR representation technique has the drawback that it represents all common subgraphs in the dynamic environment. The proposed scheme represents graph data inside each snapshot by modifying the CSR representation method. The proposed scheme modifies the CSR technique to reduce the amount of information in graph data that is duplicated in the dynamic environment and connects common subgraphs with hierarchical snapshots. Figure 6 shows a sparse

matrix using the modified CSR technique. Common subgraphs are represented as $IS_1$ and graph data except for the common subgraph are presented in $DS_1$ and $DS_2$. The CIA is modified to connect $IS_1$ and $DS_1$, $DS_2$, and the common subgraphs are connected by storing an offset in the CIA of $IS_1$ inside the lower end of $DS_1$ and $DS_2$.

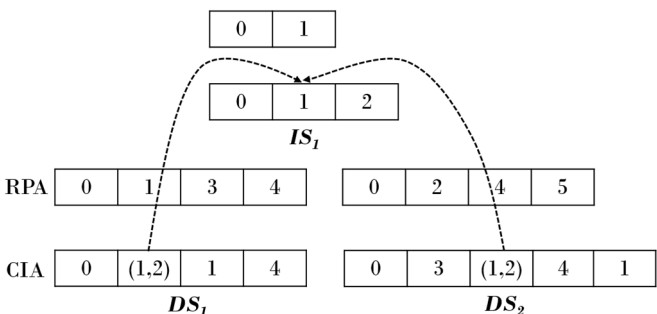

**Figure 6.** Modified compressed sparse row.

The proposed scheme can represent the information of vertices and edges in the internal structure of a snapshot. Figure 7 shows a sparse matrix that represents a single graph. Each attribute information of vertices and edges can be represented as shown in Figure 7. A large amount of unnecessary space is consumed to represent the graph data. Figure 8 shows the sparse matrix in Figure 7 represented using the modified CSR technique. In Figure 8, the ID of each vertex in the sparse matrix and the Edge Weights (EWs) that depart from vertices are represented by single intersection and DSs. The actual data in the database can be traced by utilizing ID 1124 of vertex $v_1$, and various sizes of information can be stored by utilizing the key-value type storage structure if more varied data are preferred to be inserted, as shown in the weight 0.1 of edge $e_{12}$ that is connected to the vertex.

|       | $v_1$ | $v_2$ | $v_3$ | $v_4$ | $v_5$ |
|-------|-------|-------|-------|-------|-------|
| $v_1$ | 0     | 0.1   | 0     | 0     | 0     |
| $v_2$ | 0.4   | 0     | 0.8   | 0.5   | 0     |
| $v_3$ | 0     | 0     | 0     | 0     | 0     |
| $v_4$ | 0     | 0     | 0     | 0     | 0     |
| $v_5$ | 0     | 0     | 0.9   | 0     | 0     |

**Figure 7.** Representation of graph data using a sparse matrix.

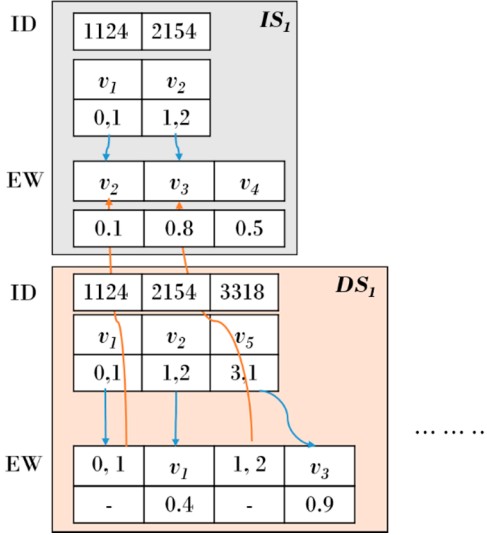

**Figure 8.** Modified CSR representation.

### 3.5. Graph Search

The graph search in the proposed scheme is more efficient when searching for histories over a certain period of time, since it manages common subgraphs in an integrated manner. For a graph search, the corresponding IS is accessed to read common graphs and DSs in the lower end to access overall graphs. Assuming that the *i*-th IS and *n*-th DS are represented by $IS_i$ and $DS_n$, a graph at time $T_n$ can be expressed by Equation (6) similar to [17]. Here, '∪' is a union operation to merge the two graphs.

$$G_n = IS_i \cup DS_n \tag{6}$$

It is an example that accesses $G_3$ through the overall structure in Figure 9. First, whether $DS_3$ is connected with $IS_2$ is verified. Once access to $IS_2$ connected to the DS at the corresponding time is complete, $DS_3$ is accessed to access each vertex in the same manner as in the IS. Since there is no new information about the creation of edges among edges connected to out-going vertex $v_1$ in the DS, offset (0, 2) of the in-coming vertex table of $IS_2$ that represents the information of edges connected with $v_1$ in the out-going vertex table of $IS_2$ is stored to read the common subgraphs. $IS_2$ refers to the offset (0, 2) that points to the in-coming vertex table $v_2$ and $v_3$ connected to $v_1$ in the out-going vertex table to represent an edge that is a connection between two vertices. Offset 0 refers to the location of the in-coming vertex table, and offset 2 refers to the number of in-coming vertices connected to out-going vertex $v_1$. In the case of the next out-going vertex $v_2$ of the DS, a new edge $e_{26}$ is generated. Thus, the in-coming vertex table pointed to by $v_2$ represents $v_6$, which indicates new connection information, and the remainder of the common information is indicated by the in-coming vertex table offset of $IS_2$ and can be accessed. Finally, all information in the DS is read through the above procedure to respond to the historical graph query at the corresponding time.

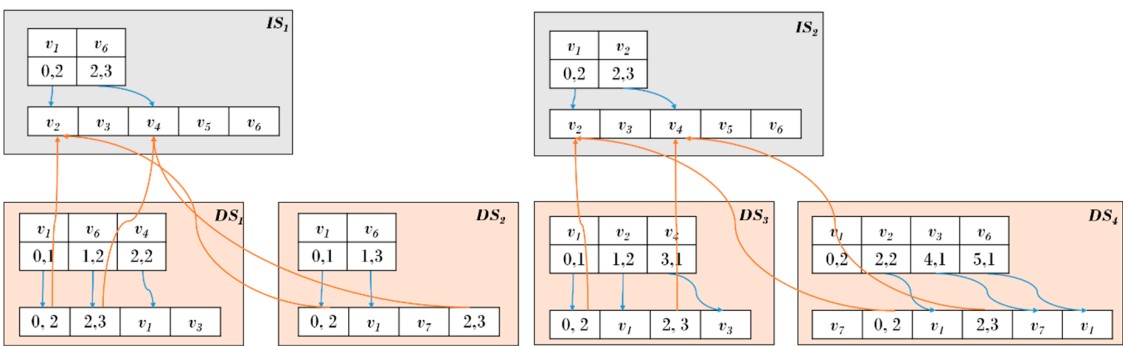

**Figure 9.** Historical graph search.

## 4. Performance Evaluation

### 4.1. Experimenral Results

The performance of the proposed scheme was verified by comparing the storage usage and query processing times with those of existing schemes. Furthermore, threshold values were also evaluated to determine which threshold showed the best efficiency. Version Graph and DeltaGraph were used as existing comparison schemes in the performance experiment. The experiment was conducted in an environment of Intel Core i5-4440 CPU 3.10 GHz with 8 GB memory. Two data sets were used for the experiment evaluation. Dataset1 is graph data that represent the monthly change in Internet service provider for 2004–2007 provided by Center for Applied Internet Data Analysis (CAIDA) [38], and Dataset2 is the historical graph data [39] generated by citation information from USA patents for 1975–1999 [40]. The data provided by CAIDA consist of approximately 25,000 vertices and 100,000 edges for temporal graphs. The citation information of USA patents consists of approximately 3,770,000 vertices and 16,510,000 edges. The history data in the citation information of USA patents is static information that includes the overall citation information. Therefore, 122 historical graphs are generated by partial modification to represent dynamic situations. The storage usage and query response times were measured to evaluate the proposed scheme's space efficiency and access speed. The proposed scheme was evaluated according to the threshold for creating an IS to verify the optimum space efficiency and compared with existing schemes for history graph retrieval.

Figure 10 compares the amount of storage space used by IS generation in the proposed scheme. The proposed scheme uses the CSGR threshold value in the changed graph to generate IS. The candidate threshold values were set to 0.5, 0.6, and 0.7 to generate the IS. If the CSGR threshold is small, the size of the subgraph stored in the DS increases, although many ISs are not generated. Conversely, large CSGR thresholds result in a large number of ISs being generated, and the size of the subgraphs stored in the DS is reduced. Since Dataset1 does not have many vertices and edges that make up the graph, there is not much change in the size of the storage space according to the CSGR threshold. However, since Dataset2 contains many vertices and edges compared to Dataset1, there is a lot of variation in storage space with changes in CSGR thresholds. Storage space usage was the highest when the CSGR threshold was 0.7, although there were differences in relative performance depending on data characteristics. The space efficiency was improved by up to 28% when using 0.6 of the threshold compared to the other threshold values.

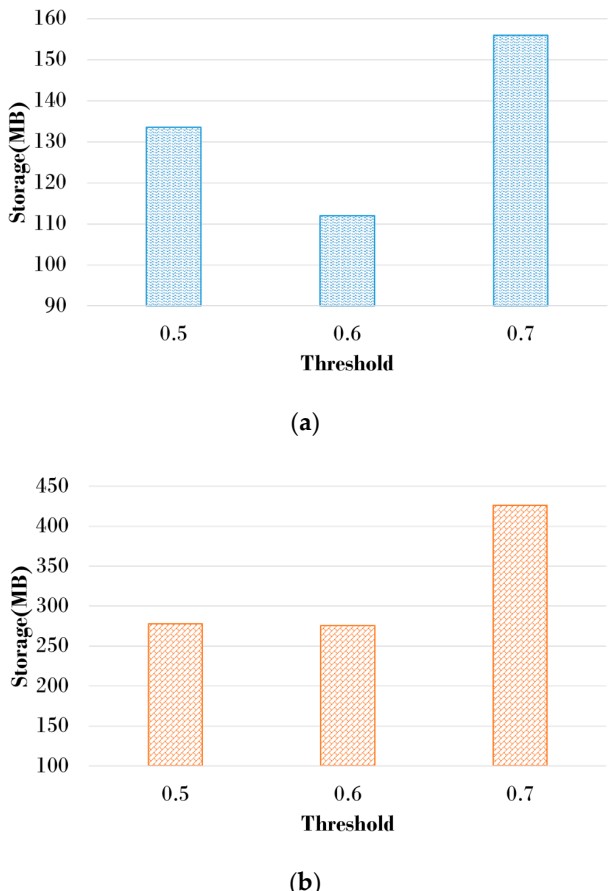

(**a**)

(**b**)

**Figure 10.** Storage usage according to the threshold: (**a**) Dataset1; (**b**) Dataset2.

The proposed scheme reduces the size of storage space using the modified CSR when managing historical graphs through IS and DS. Performance evaluation is performed while changing the CSGR threshold to check the size change of storage space according to the application of the modified CSR technique. In Figure 10, when the CSGR threshold was set to 0.7, the relative performance improvement resulting from the application of CSR was the best because storage space usage was the highest. Similar to Figure 10, when the CSGR threshold was set to 0.6, the overall storage space usage was the smallest. The space efficiency was improved by up to 51% when using modified CSR compared to the proposed scheme without modified CSR, as shown in Figure 11. This result showed that the amount of information is reduced by using modified CSR.

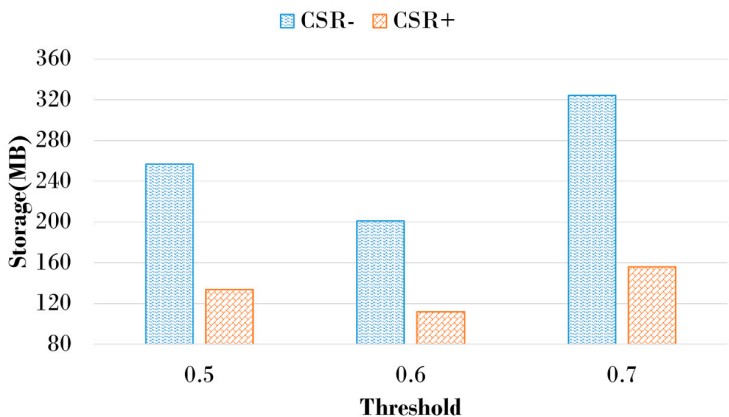

**Figure 11.** Storage usages using or without modified CSR.

Figure 12 shows the query processing time according to the CSGR threshold. Depending on the CSGR threshold, the size of IS and DS was different. To examine the search efficiency for historical graphs, the query processing time is compared, creating a graph storage structure according to CSGR thresholds and changing the time range to search. Five query types were created to search for graphs that exist within a specific time range for historical graphs that have changed on a monthly basis. As a result of comparing query processing time while changing the time range of the graph to be searched, the processing time increases relatively as the time range increases. When the query time range was 7, the query processing time was the smallest, but the relative performance according to the CSGR threshold was the largest. This indicates that IS and DS produced by CSGR thresholds have a significant impact on performance when the time range to search is small. A large query time ranges requires access to historical graphs existing in many time ranges, so the relative performance differences between CSGR thresholds are not significant. The best performance was found at 0.6 of the threshold. A smaller CSGR threshold value increased the number of DSs connected to the IS, which decreased the number of ISs. Since the number of ISs was small, the time required to access the ISs was decreased, thereby showing better performance. Using a large threshold value generated a large number of ISs thereby increasing the time required to access the ISs, leading to an increased query processing time. It is necessary to select an appropriate threshold value that considers both space efficiency and retrieval performance.

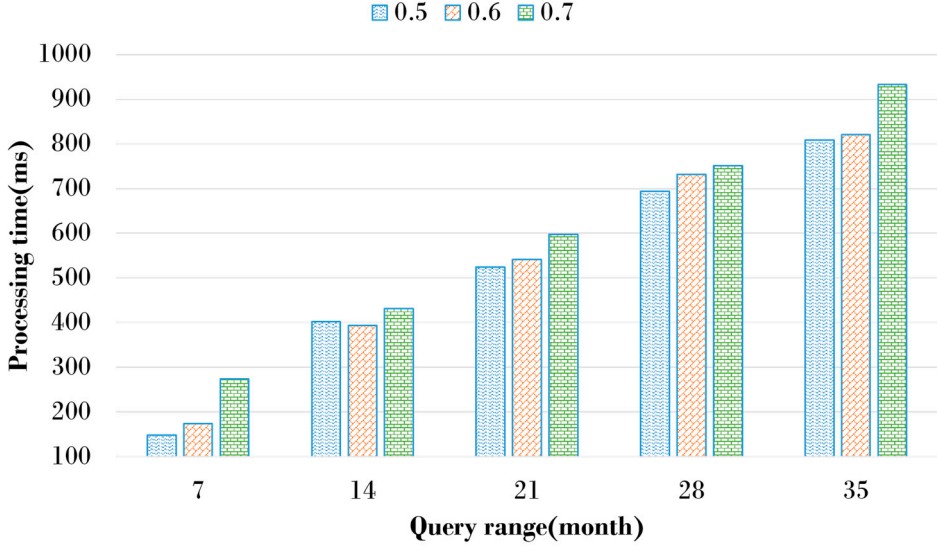

**Figure 12.** Processing time according to threshold.

The proposed scheme is designed to increase the efficiency of history searches while reducing the storage space on the history graph. In order to demonstrate the excellence of the proposed scheme, we compare it with the existing scheme in terms of the size of storage space. Figure 13 shows the storage space of the existing schemes and the proposed scheme. Since the efficiency of storage space is best when the CSGR threshold is 0.6 in its own performance evaluation of the proposed technique, the proposed technique in this experiment compared the size of the storage space with the existing technique after setting the CSGR threshold to 0.6. Since the efficiency of storage space is the best when the CSGR threshold is 0.6 in its own performance evaluation of the proposed scheme, we compare it with the existing schemes in terms of the size of the storage space after setting the CSGR threshold to 0.6. Since Dataset1 has relatively few vertices and edges that make up the graph compared to Dataset2, the space for storing historical graphs is small compared to Dataset2. Version Graph takes up the largest storage space because it stores information on all vertices and edges that have changed over time. However, DeltaGraph and the proposed scheme take up a relatively small amount of space

because it stores only the changed information based on the baseline snapshot graph. DeltaGraph uses the tree structure to manage the baseline snapshot graph and to manage the changes made to the baseline snapshot graph as an event list. However, the proposed scheme uses less storage space than DeltaGraph because IS is generated using CSGR, the ratio of common graphs that have not changed over time, and only changes are stored in DS. In addition, the storage space is the smallest because it compresses and stores graphs using CSR. The results of two experiments showed that Version Graph maintained the structures of all graphs and the life information of all elements, and approximately 80% of space waste was consumed compared to the proposed scheme. Since DeltaGraph has to store a large amount of history data to track the history, it consumed approximately 47% more space than that of the proposed scheme. The proposed scheme utilized common subgraphs efficiently, which revealed a better performance than comparable schemes.

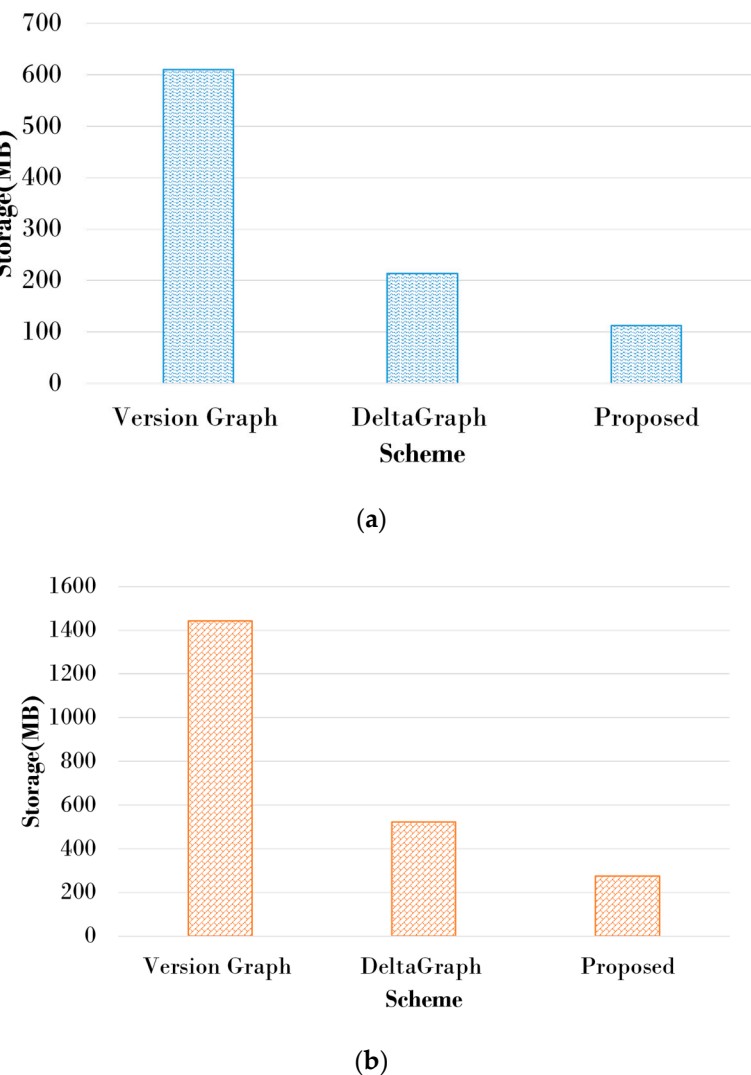

**Figure 13.** Storage usage comparison: (**a**) Dataset1; (**b**) Dataset2.

Historical graphs should maintain changed graphs from the past to the present and be able to search past graphs. Figure 14 shows the query processing time of the proposed scheme and the existing schemes. To evaluate the search efficiency for historical graphs, the proposed scheme sets the CSGR threshold to 0.6 and compares the query processing time with the existing schemes. Five query types were created to search for graphs that exist within a specific time range for historical graphs that have changed on a monthly basis. Since Version Graph stores information about all vertices and edges that

have changed over time, many vertices and edges should be compared even if the search time range is small. Therefore, Version Graph takes more query processing time than DeltaGraph and the proposed scheme, even if the search time range is small, but even if the search time range is changed, the query processing time does not change much. Since DeltaGraph stores historical graphs in a tree structure, the query processing time is the fastest when the search time range is small. However, DeltaGraph creates a graph by comparing the list of events stored on the leaf node based on a snapshot, so the query processing time decreases significantly as the search time range increases. The proposed scheme stores unchanged common subgraphs on the changed graph in IS and only actual modified subgraphs in DS. Therefore, because a baseline snapshot is created based on the ratio of changes in the graph, smaller search time ranges take relatively more time to process queries than DeltaGraph, but as search time increases, there is less time to process queries than the existing schemes. The performance evaluation showed that DeltaGraph had the best performance if the search time range was small, but Version Graph showed the best performance as the time range increased. The proposed scheme has lower query processing performance than DeltaGraph if the time range to search is small, but as the time range increases, it shows better performance than the existing schemes. This is because the proposed scheme manages common subgraphs using the change rate of the graph, so it does not take much time to search for the changed subgraph.

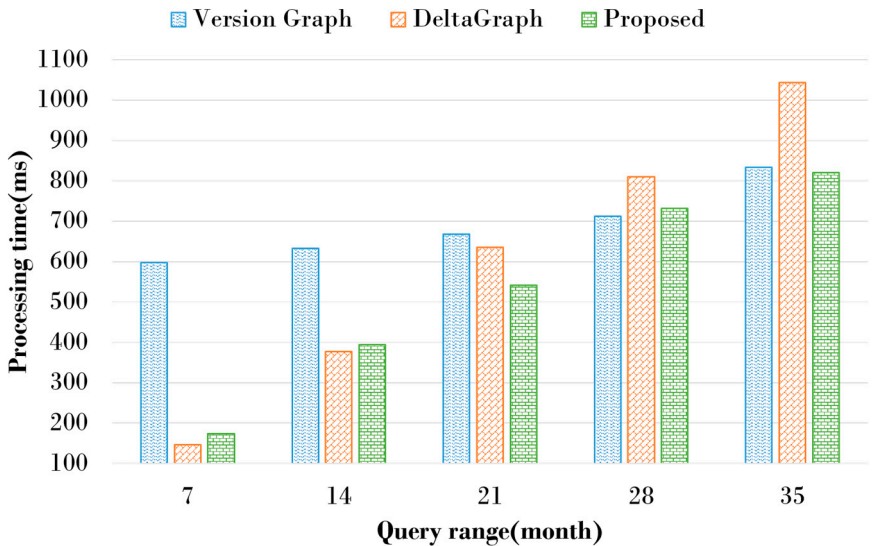

**Figure 14.** Query processing time according to the query range.

### 4.2. Discussion

Various historical graph management schemes have been proposed to track and manage the history of changes in dynamic graphs with continuous changes. The typical historical graph management schemes are Version Graph and DeltaGraph. We analyze and compare the existing schemes and the proposed scheme in terms of storage space and query processing time. Version Graph records lifespan information to determine the existence of vertices and edges at a specific time to maintain the structure of all graphs that have changed in the past. The lifespan is expressed as the time interval in which vertices and edges exists. Therefore, the size of the Version Graph can continue to increase if the graph changes a lot, and the lifespan information stored on the graph takes up a lot of space, especially if insertion and deletion are repeated. DeltaGraph manages the history of changes in graph over time in a tree structure. The leaf nodes in the tree structure store the changes by time zone based on specific snapshot graphs as events. The proposed scheme manages changes made in chronological order through the provenance model and generates IS using CSGR when many changes in the graph occur. Therefore, the proposed scheme creates IS if there are many changes to the graph and stores only the

changed subgraphs in the DS. It also reduces historical graph storage space by compressing and storing graphs through the modified CSR technique. As a result of the performance evaluation, DeltaGraph uses less storage space than Version Graph because it stores only the changed event lists based on the baseline snapshot. The proposed scheme uses the smallest space because it not only reduces the number of subgraphs that are duplicated through CSGR but also compresses the graph through the modified CSR.

Historical graphs should be able to store graph history changes and search for graphs that have changed from the past to the present. Version Graph expresses the changes in the subgraphs in a hierarchical structure and searches for graph changes over time using SCC. However, since it represents changes in the overall graph as a single integrated model, searching for a graph that exists at a specific time takes a lot of query processing time because it checks the lifespan for the entire graph. DeltaGraph searches the baseline snapshot graph through the tree structure, sequentially approaches the leaf nodes, and generates the graph using the event list. Therefore, since DeltaGraph provides quick access to a baseline snapshot graph through a hierarchical structure, it has the best query processing performance when the time range to search is small. However, DeltaGraph creates a graph while traversing the event list stored on the leaf node based on the baseline snapshot graph, so as the time range to search increases, the query processing performance decreases significantly. The proposed scheme can store common subgraphs in IS, store changed subgraphs in DS, and connect IS and DS to search history subgraphs. Since the proposed scheme does not have a baseline snapshot structure such as DeltaGraph, smaller time ranges to search result in lower query processing performance than DeltaGraph. However, since the proposed scheme creates IS by taking account of the graph change rate through CSGR, if there are many changes in the graph, it creates a new IS and stores only the changed subgraphs in the DS. Therefore, the proposed scheme improves query processing performance over DeltaGraph as the time range to search increases. Performance evaluations show that VERSION GRAPH has little change in query processing performance when the time range to search is small, but with increased time range to search graph history, there is little change in query processing performance. The proposed scheme shows the best performance when the time range to search is smaller than DeltaGraph, but when the time range to search is large, the query processing performance is lower than that of DeltaGraph.

## 5. Conclusions

In this paper, we proposed a graph management scheme that can effectively store and search for changes in graphs that have changed from the past. The proposed scheme manages the graph by dividing it into IS and DS using the characteristics that changes in the graph over time occur only at some vertices and edges, and the ratio of changes in the graph. As a baseline snapshot graph, IS stores common subgraphs that have not been changed, while DS stores only subgraphs that have been changed based on IS. When a graph is changed, a modified graph history is managed by integrating it into one graph using the provenance model, and a new IS is created when the graph changes above the threshold using CSGR. This resolves the problem of redundant storage of unchanged subgraphs. In addition, the proposed scheme can reduce storage space on the historical graphs because it stores them by transforming the CSR that compresses and stores the graph. The query performance can be improved by only accessing change histories except for common subgraphs during a historical graph search. The experiment results showed that the proposed scheme improved space efficiency by up to 80% compared to existing schemes. However, it cannot guarantee better performance for a small range of queries, even though it improved the query processing time by approximately 20% compared to existing schemes for a large range of queries. In the near future, we will design an index structure to improve the performance of history searches.

**Author Contributions:** Conceptualization, K.B., G.K. and J.Y.; methodology K.B., G.K. and J.Y.; software, G.K.; validation, K.B., J.L. and G.K.; formal analysis, K.B., G.K. and J.Y.; data curation, G.K. and J.L.; writing—original draft preparation, K.B. and G.K.; writing—review and editing, J.L. and J.Y. All authors have read and agreed to the published version of the manuscript.

**Funding:** This work was supported by the National Research Foundation of Korea (NRF) grant funded by the Korea government (MSIT) (No. 2019R1A2C2084257), by Next-Generation Information Computing Development Program through the National Research Foundation of Korea (NRF) funded by the Ministry of Science, ICT (No. NRF-2017M3C4A7069432), by the Ministry of Education of the Republic of Korea and the National Research Foundation of Korea (NRF-2017S1A5B8059946), and by Institute of Information & Communications Technology Planning & Evaluation (IITP) grant funded by the Korea government (MSIT) (No.B0101-15-0266, Development of High Performance Visual BigData Discovery Platform for Large-Scale Realtime Data Analysis).

**Conflicts of Interest:** The authors declare no conflict of interest.

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
