# Peer review of "Historical Graph Management in Dynamic Environments"

_electronics, doi:10.3390/electronics9060895_

Round 1
Reviewer 1 Report
The proposed work presents a new historical graph management scheme which consists of an intersection snapshot and a delta snapshot to enhance storage utilization and an accessibility of a snapshot graph. Claimed scheme constantly detects graph change and calculates a common subgraph ratio between historical graphs over time. The superiority of the proposed scheme is provided through various performance evaluations as well.
The proposed work is interesting and within the scope of the journal. The superiority of the presented work is well presented. Related work is well explained and update. I have some minor concerns which are as follows:
(1) Please improve the plots and especially fonts within the plot as they are not clearly visible when printed.
(2) Provide references for the equations which are taken from literature.
(3) All acronyms should be defined at first appearance.
(4) Paper needs to be double-checked for typos and grammatical errors.
Author Response
We would like to sincerely thank you for your attentive indications and good comments. Our paper is partially rewritten in order to revise and complement your comments.
Please refer to the attached file.

Reviewer 2 Report
The work done is interesting, well presented, and in general, presents a good originality and proposal both methodological and practical.
However, I believe that there are some critical aspects that should be resolved before they can be published:
- The introduction makes a somewhat strange initial motivation and argumentation, lacking references or quotations to support such statements, and without actually introducing the objective, scope, methodology or approach followed.
- The theoretical framework section requires a much greater effort to show the state of the current issue, as well as related works, with its quotations and detailed explanations. This part suffers from a complete systematic review of solutions similar to the one proposed by the authors. In general, I believe that this systematic review of other similar proposals should also be used as a common thread in the evaluation, comparing the results in specific scenarios obtained by the authors' proposal and others existing in the academic world.
- The evaluation should include a much more detailed description of the applied methodology,
- It would include a discussion section on the threats to the validation carried out (internal, external threats, etc.) in order to increase the validity of this section.
- The conclusions are too brief. I think the authors can improve this part
Author Response

(The authors gave the same response as above.)

Reviewer 3 Report
The manuscript presents a novel methodological approach to derive and manage historical graphs. It includes Intersection and Delta Snapshots.
The paper is well written. The methodological approach is clearly structured and presented in a transparent way. The outcomes are visually summarized in Figure 14 which shows differences in the processing time between three methods (Version Graph, Delta Graph and the proposed method).
There are only minor points which I would like to mention. I would like to ask the authors to consider the following two points in a final version of the manuscript:
- The term historical graph leads to misunderstandings. It could be understood as a graph representing historical developments. Could you please define what is meant by historical data in the context of your research? You refer to several related studies [12-15) in line 33. This could be the passage where you insert a definition of this term.
- It would be much nicer for readers (including readers of other disciplines) to get to know application scenarios of your research. Which applicational fields of society would benefit from an improved data representation through graphs? I could imagine that the field of Web Cartography would be a good example to illustrate the improvement potential. In these days of the COVID-19 crisis, a lot of web cartographic applications are available worldwide that show the spreading of the virus spatio-temporally in interactive maps. Graphs are an established cartographic method to represent data in maps. It could be an idea to introduce the application field of cartography and refer to examples that emphasize the recent methodological developments of online available web cartography and spatial data representation. Examples are:
Edler, D. & Vetter, M. (2019). The Simplicity of Modern Audiovisual Web Cartography: An Example with the Open-Source JavaScript Library leaflet.js. In: KN – Journal of Cartography and Geographic Information, 69 (1): 51-62. https://doi.org/10.1007/s42489-019-00006-2
Horbinski, T. & Lorek, D. (2020). The use of Leaflet and GeoJSON files for creating the inter-active web map of the preindustrial state of the natural environment. Journal of Spatial Science, online first: https://doi.org/10.1080/14498596.2020.1713237
Author Response

(The authors gave the same response as above.)

Round 2
Reviewer 2 Report
The authors have done a good job of reviewing, improving many of the aspects that we commented on for correction. Some aspects can still be improved but I think the work done with the review is adequate.